# RNA structures that resist degradation by Xrn1 produce a pathogenic Dengue virus RNA

Erich G Chapman[1], Stephanie L Moon[2], Jeffrey Wilusz[2], Jeffrey S Kieft[1]*

[1]Department of Biochemistry and Molecular Genetics, Howard Hughes Medical Institute, University of Colorado Denver School of Medicine, Aurora, United States; [2]Department of Microbiology, Immunology and Pathology, Colorado State University, Fort Collins, United States

**Abstract** Dengue virus is a growing global health threat. Dengue and other flaviviruses commandeer the host cell's RNA degradation machinery to generate the small flaviviral RNA (sfRNA), a noncoding RNA that induces cytopathicity and pathogenesis. Host cell exonuclease Xrn1 likely loads on the 5′ end of viral genomic RNA and degrades processively through ~10 kB of RNA, halting near the 3′ end of the viral RNA. The surviving RNA is the sfRNA. We interrogated the architecture of the complete Dengue 2 sfRNA, identifying five independently-folded RNA structures, two of which quantitatively confer Xrn1 resistance. We developed an assay for real-time monitoring of Xrn1 resistance that we used with mutagenesis and RNA folding experiments to show that Xrn1-resistant RNAs adopt a specific fold organized around a three-way junction. Disrupting the junction's fold eliminates the buildup of disease-related sfRNAs in human cells infected with a flavivirus, directly linking RNA structure to sfRNA production.

*For correspondence: jeffrey.
kieft@ucdenver.edu

**Competing interests:** The authors declare that no competing interests exist.

## Introduction

Flaviviruses (FVs) are single-stranded, (+)-sense RNA viruses that include West Nile (WNV), Yellow Fever (YFV), Japanese Encephalitis (JEV) and other human disease-causing viruses (*Fields et al., 2013*). Dengue (DENV), the most pervasive FV, was the cause of >100 million symptomatic human infections during 2010 (*Bhatt et al., 2013*). In humans, DENV infection can lead to severe hemorrhagic fever, dengue shock syndrome, and death (*Simmons and Farrar, 2012*; *The World Health Organization, 2014*; *United States Centers for Disease Control and Prevention, 2014*). Over 40% of the world's population is at risk of contracting this disease, (*The World Health Organization, 2014*; *United States Centers for Disease Control and Prevention, 2014*) and global trade and climate change continue to extend the habitat of the mosquito vector that spreads the virus (*Barclay, 2008*; *Morin et al., 2013*; *Murray et al., 2013*; *Brown et al., 2014*). There is no broadly-effective therapy or vaccine against DENV or many other FVs, motivating continued research into the molecular basis of disease to identify new vulnerabilities in the viral lifecycle.

FVs enter the cell through receptor-mediated endocytosis, culminating in the release of an infectious, ~10.5 kB genomic RNA (gRNA) into the cytoplasm (*Kuhn et al., 2002*). The capped but non-polyadenylated viral genomic RNA (gRNA) encodes a single open reading frame (ORF), flanked by structured 5′ and 3′ untranslated regions (UTRs). Translation of the FV ORF produces a single polypeptide that is processed by both viral and cellular proteases to yield ten viral proteins (*Fields et al., 2013*). Both FV UTRs fulfill critical roles in the viral lifecycle, (*Iglesias and Gamarnik, 2011*) including forming base pairing interactions during (−) strand synthesis of the viral RNA (*Filomatori et al., 2006*; *Villordo and Gamarnik, 2009*; *Friebe and Harris, 2010*; *Villordo et al., 2010*; *Gebhard et al., 2011*).

**eLife digest** More than 40% of people around the globe are at risk of being bitten by mosquitoes infected with the virus that causes Dengue fever. Every year, more than 100 million of these individuals are infected. Many develop severe headaches, pain, and fever, but some develop a life-threatening condition where tiny blood vessels in the body begin to leak. If not treated quickly, this more severe manifestation of the illness can lead to death.

There are currently no specific therapies or vaccines against Dengue or many other closely related viruses such as West Nile and Japanese Encephalitis. These viruses use instructions encoded in a single strand of RNA to take over an infected cell and to reproduce. The viruses also exploit an enzyme that cells use to destroy RNA to instead produce short stretches of RNA called sfRNAs that, among other things, may help the virus to avoid the immune system of its host. Understanding exactly how Dengue and other viruses thwart this enzyme—which is called Xrn1—may help scientists develop treatments or vaccines for these diseases.

Chapman et al. have now shown that Dengue virus RNA contains a number of RNA elements that prevent it being completely degraded by the Xrn1 enzyme. In particular, a junction formed by three RNA helixes is critical for stopping the enzyme in its tracks, leaving the disease-associated sfRNA behind. A single mutation in the Dengue RNA disrupts the structure of the three-helix junction and allows the enzyme to completely destroy the RNA. A similar mutation was also made in the West Nile virus RNA and when human cells were infected with the mutated West Nile virus, the short sfRNAs were not produced. Treatments or vaccines targeting this structure may therefore help reduce illness associated with Dengue and related viruses.

In addition to replicated copies of the gRNA, high levels of shorter non-coding viral RNAs are also produced during FV infection (*Naeve and Trent, 1978*; *Takeda et al., 1978*; *Wengler and Gross, 1978*; *Urosevic et al., 1997*). These subgenomic flaviviral RNAs (sfRNAs) contain most of the 3′UTR of the FV gRNA and play an important role in regulating the switch between translation and replication of the viral genome (*Lin et al., 2004*). Production of sfRNAs is an evolutionarily-conserved trait in arthropod-borne FV's and its accumulation is directly linked to human disease (*Pijlman et al., 2008*). Importantly, through still poorly-understood mechanisms, sfRNA controls WNV's ability to evoke cytopathicity in mammalian cell culture and the onset of pathogenesis in fetal mice (*Pijlman et al., 2008*; *Liu et al., 2014*). Virological studies of sfRNAs from WNV, DENV (*Liu et al., 2010*) and other flaviviruses showed that they disrupt aspects of the immune response, affecting RNAi mechanisms (*Schnettler et al., 2012*), mRNA turnover pathways (*Moon et al., 2012*), and the type-I interferon response (*Schuessler et al., 2012*; *Chang et al., 2013*). The importance of sfRNAs in FV-induced disease raised questions about the mechanism by which these RNAs are produced and maintained at levels approaching or exceeding those of viral gRNA (*Fan et al., 2011*). Several studies showed that sfRNAs form by viral manipulation of the host cell's RNA turnover machinery. Specifically, sfRNAs form as the result of incomplete degradation of viral gRNA by the cytoplasmic exoribonuclease Xrn1 (*Funk et al., 2010*; *Silva et al., 2010*). Xrn1 is a processive, 5′→3′ exonuclease responsible for degradation of roughly 30–40% of mRNA in actively dividing cells (*Jones et al., 2012*). sfRNA production occurs when viral gRNA is loaded into Xrn1 for decay. Xrn1 degrades through nearly the entire viral gRNA before stalling near the beginning of the 3′UTR; what remains is an sfRNA (*Figure 1A*).

Examination of Xrn1 illustrates how remarkable it is that FVs have evolved a way to resist the exonuclease at a specific and defined point within the gRNA. RNA degradation by Xrn1 takes place in the enzyme's catalytic chamber buried within a conserved active site (*Chang et al., 2011*; *Jinek et al., 2011*). Progression of Xrn1 along an RNA being degraded is hypothesized to be powered by a combination of electrostatic interactions specific for 5′-monophosphorylated RNAs and by repetitious π-stacking of RNA nucleobases with aromatic residues in the enzyme's active site (*Jinek et al., 2011*). These features enable Xrn1 to degrade through complex RNA structures, including decapped mRNAs and ribosomal RNA (rRNA). Thus, a popular practical application for Xrn1 is eliminating unwanted monophosphorylated RNA (e.g., rRNAs) from samples bound for next-generation sequencing.

How do sfRNAs escape degradation by Xrn1? Current evidence suggests that specific RNA sequences and structures located in the 3′UTR of different FVs are the signals that cause Xrn1 to stall.

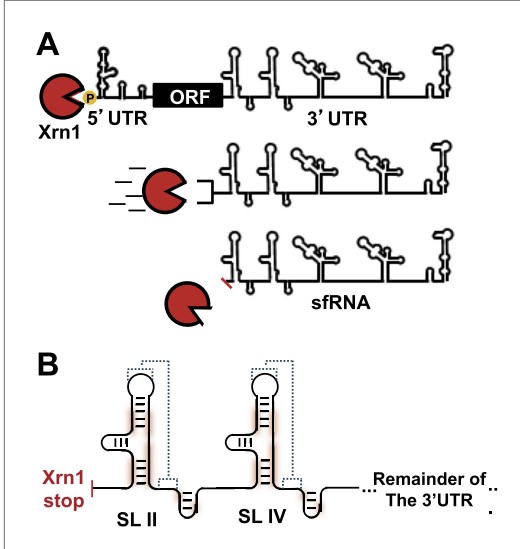

**Figure 1**. sfRNAs are formed by incomplete degradation of the flaviviral genomic RNA by the 5′→3′ exonuclease Xrn1. (**A**) Xrn1 (red) likely loads onto decapped, monophosphorylated gRNA and degrades ~10 kb of RNA before stopping within the viral 3′UTR. The remaining RNA is the sfRNA. (**B**) In the sfRNAs studied to date, stem-loop (SL) elements near the 5′ border of the 3′UTR/sfRNA appear to be signals for Xrn1 resistance, and are depicted here as cartoon secondary structures. In Dengue and many other FVs, two of these SLs are present in tandem, as shown. The most highly-conserved parts of the RNA are shaded red and putative PK interactions are indicated with dashed lines. Dumbbell elements (not shown) are located 3′ to these SLs.

Presumably, these RNAs have unique features that make them impervious to Xrn1. It has been proposed that both stem-loop (SL) and dumbbell (DB) type structures found in the FVs 3′UTRs can halt Xrn1 progression (*Figure 1B*; *Pijlman et al., 2008*; *Funk et al., 2010*; *Silva et al., 2010*). Of these putatively resistant structures, the SL elements from WNV (*Funk et al., 2010*) and YFV (*Silva et al., 2010*) have been explored. Phylogenetic studies identified ~19 conserved nucleotides within a secondary structure predicted to include a three-helix junction, a downstream hairpin, and an RNA pseudoknot (PK) (*Pijlman et al., 2008*). The presence of the PK is supported by virological studies (*Silva et al., 2010*), and this led to the proposal that these are 'rigid' structures. However, our current understanding of these RNA structures does not explain their ability to halt Xrn1 and in the case of DENV, the relationship of the structure of the 3′UTR to its Xrn1-resistant properties is unexplored.

Here, we describe our interrogation of the structure and Xrn1-resistant properties of the complete 3′UTR of a serotype 2 Dengue Virus (DENV2). Chemical probing of the global structure of the 3′UTR revealed five independently-folded RNA structures within the longer RNA. We recapitulated Xrn1 resistance in vitro and developed a new fluorescence-based assay capable of quantitative, real-time monitoring of RNA degradation by Xrn1 and resistance to the enzyme. We used this assay to explore the Xrn1-resistant behavior of RNA structures throughout the DENV2 3′UTR, identifying two Xrn1-resistant RNA structures that confer quantitative protection to downstream segments of RNA. Using one of these structures we developed a series of mutants and precisely mapped where Xrn1 halts, revealing that a specifically-structured three-way junction is a functionally critical element of Xrn1 resistance. Similar mutations within the Xrn1-resistant structures of another FV (the Kunjin strain of WNV), also impair the formation of sfRNAs in vitro and in infected human cells. Cumulatively, these results set the stage for detailed mechanistic and structural studies of these unique, disease-related viral RNAs.

## Results

### Structural architecture of the complete DENV2 3′UTR

We first characterized the global architecture of the DENV2 3′UTR and the secondary structures of five shorter RNA sequences within the UTR that are predicted to form discrete secondary structure domains (*Figure 2—figure supplement 1*). We probed the structure of these RNAs using dimethyl sulfate (DMS) and N-methylisatoic anhydride (NMIA, SHAPE chemistry) (*Tijerina et al., 2007*; *Weeks, 2010*; *McGinnis et al., 2012*), and used the resultant data to produce an experimentally-supported secondary structure of the complete 3′UTR (*Figure 2A,B*). The chemical reactivity profiles obtained by probing individual regions of the 3′UTR in isolation match the profiles of those regions in the context of the intact 3′UTR (*Figure 2C*). This correlation suggests that individual regions of the 3′UTR fold the same when in isolation as they do in the context of the intact 3′UTR, and thus can be studied as structurally discrete elements. The resultant secondary structure model is largely consistent with previous *in silico* predictions (*Hahn et al., 1987*; *Proutski et al., 1997*; *Olsthoorn and Bol, 2001*). Two stem-loop

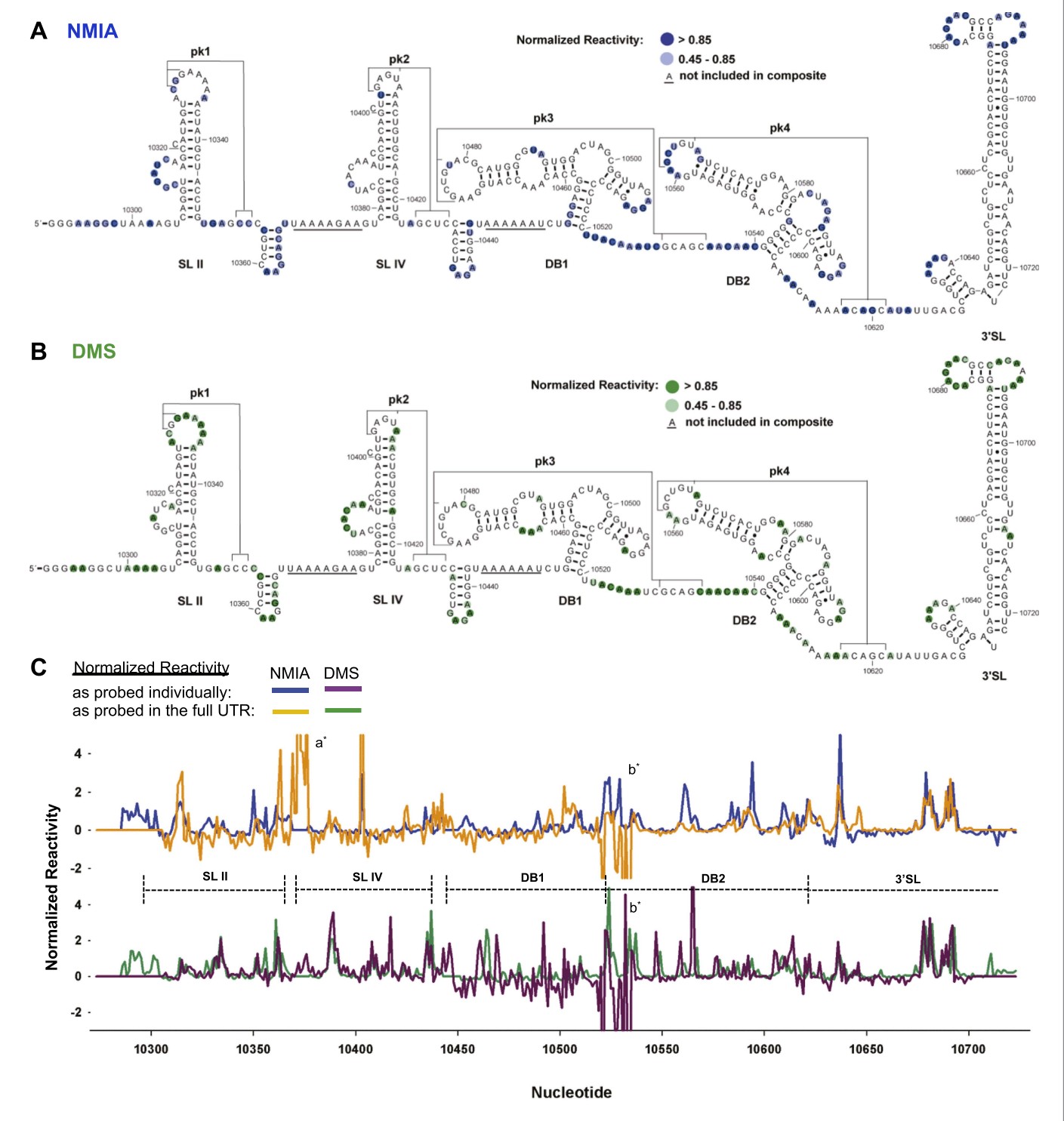

**Figure 2**. Chemical probing and predicted secondary structure of the DENV2 3′UTR. (**A**) The Shapeknots predicted secondary structure of the DENV2 3′UTR with NMIA reactivity data overlaid as indicated (blue). Secondary structure elements are labeled. (**B**) Same as panel (**A**), but with DMS reactivity data overlaid (green). (**C**) Chemical reactivity profiles of the DENV 3′UTR obtained when it is mapped in its entirety or a series of individually transcribed domains. The nucleotide position/number is on the x-axis, the y-axis is normalized reactivity. Locations of secondary structure elements are shown with dashed lines. An ' * ' indicates a region where the reverse transcriptase tended to stop in the full length 3′UTR. See **Supporting Information** for additional details.

*Figure 2. Continued on next page*

*Figure 2. Continued*

The following figure supplements are available for figure 2:

**Figure supplement 1**. Map of the RNAs used during the chemical probing experiments described in this work.

domains (SL II and SL IV) are followed by two 'dumbbell' motifs (DB1 and DB2) and by an extended 3' stem-loop structure (3'SL) that is conserved among FVs. Each of these elements has distinct features:

### Analysis of SL II

Nucleotides 10,273–10,368 are predicted to contain an Xrn1-resistant RNA structure. When input in the RNA secondary structure prediction program Shapeknots, our probing data predict a structure containing a stem-loop with a large bulge (*Figure 2A,B*). In other FVs this bulge could form a variable-length helix that emerges from a three-helix junction and formed by conserved sequence elements (*Figure 3A*; *Pijlman et al., 2008*). Considering the presence of this helix in other FV's, we propose a refined secondary structure that is consistent with both our chemical probing data and phylogenetic conservation (*Figure 3B,C*). We have used standard RNA secondary structure naming conventions to designate elements of the structure (*Figure 3A*). Based on their homology to sequences in other FV's, several nucleotides in L1 (10,329–10,330) and just downstream of P1 (10,353–10,354) are predicted to participate in the formation of an RNA pseudoknot (PK). In our experiments these positions react with both NMIA and DMS (*Figure 3B,C*) arguing against the formation of a stable PK. If a PK does form, it may be transient under these conditions. Similar reactivity was recently observed by others in a related study (*Sztuba-Solinska et al., 2013*).

### Analysis of SL IV

Nucleotides 10,358–10,443 are predicted to form a second Xrn1-resistant RNA structure. Accordingly, our chemical probing data predict a secondary structure similar to that of SL II except nucleotides located in the L3 apical loop (nucleotides 10,404–10,407, 5'-GAGU-3') and immediately downstream of helix P1 (nucleotides 10,425–10,428, 5-GCUC-3') are unreactive to both NMIA and DMS. These data suggest the formation of a stable PK that is predicted by Shapeknots and is therefore included in the refined secondary structure (*Figure 4*).

### Analysis of DB1 and DB2

The DENV2 3'UTR is predicted to contain two adjacent 'dumbbell' motifs spanning nucleotides 10,450–10,535 (DB1) and 10,541–10,624 (DB2) (*Proutski et al., 1997*; *Manzano et al., 2011*). Similar to SL II and SL IV, nucleotides located within the apical loops of these structures are predicted to form PK interactions with downstream sequence elements. Correspondingly, the chemical probing suggests the formation of a PK in DB1, where a lack of chemical reactivity suggests nucleotides 10,474–10,478 base pair with nucleotides 10,530–10,534. In contrast, in DB2 the analogous positions (nucleotides 10,562–10,566 and 10,617–10,621) react with NMIA, arguing against the formation of a similarly stable PK. Interestingly, this difference between the two DB's matches with observations that these structures fulfill different requirements in the FV lifecycle (*Manzano et al., 2011*).

### Analysis of the 3'SL

Our chemical probing data are consistent with the formation of a previously-identified stem-loop structure and small upstream hairpin at the extreme 3' terminus of the viral genome (nucleotides 10,629–10,723) (*Proutski et al., 1997*). In our experiments we do not observe evidence for a previously-suggested PK interaction between the two sub-domains (*Shi et al., 1996*), however such a structure may be transient or unstable under the probing conditions.

## Recapitulating Xrn1 resistance in vitro

We next sought to determine the ability of each of the aforementioned discrete RNA structure from the DENV2 3'UTR to resist degradation by Xrn1. These experiments were motivated in part by evidence that multiple Xrn1-resistant structures may be contained within a single FV 3'UTR (*Funk et al., 2010*; *Silva et al., 2010*). We therefore developed an in vitro assay capable of reporting Xrn1 resistance using RNAs transcribed with a 31 nucleotide 'leader' upstream of a putatively Xrn1-resistant structure. These RNAs were treated with a bacterial RNA pyrophosphate hydrolase from *Bedelovibrio bacteriovorus* (BdRppH) to convert the 5'-triphosphate generated during in vitro transcription to a monophosphate

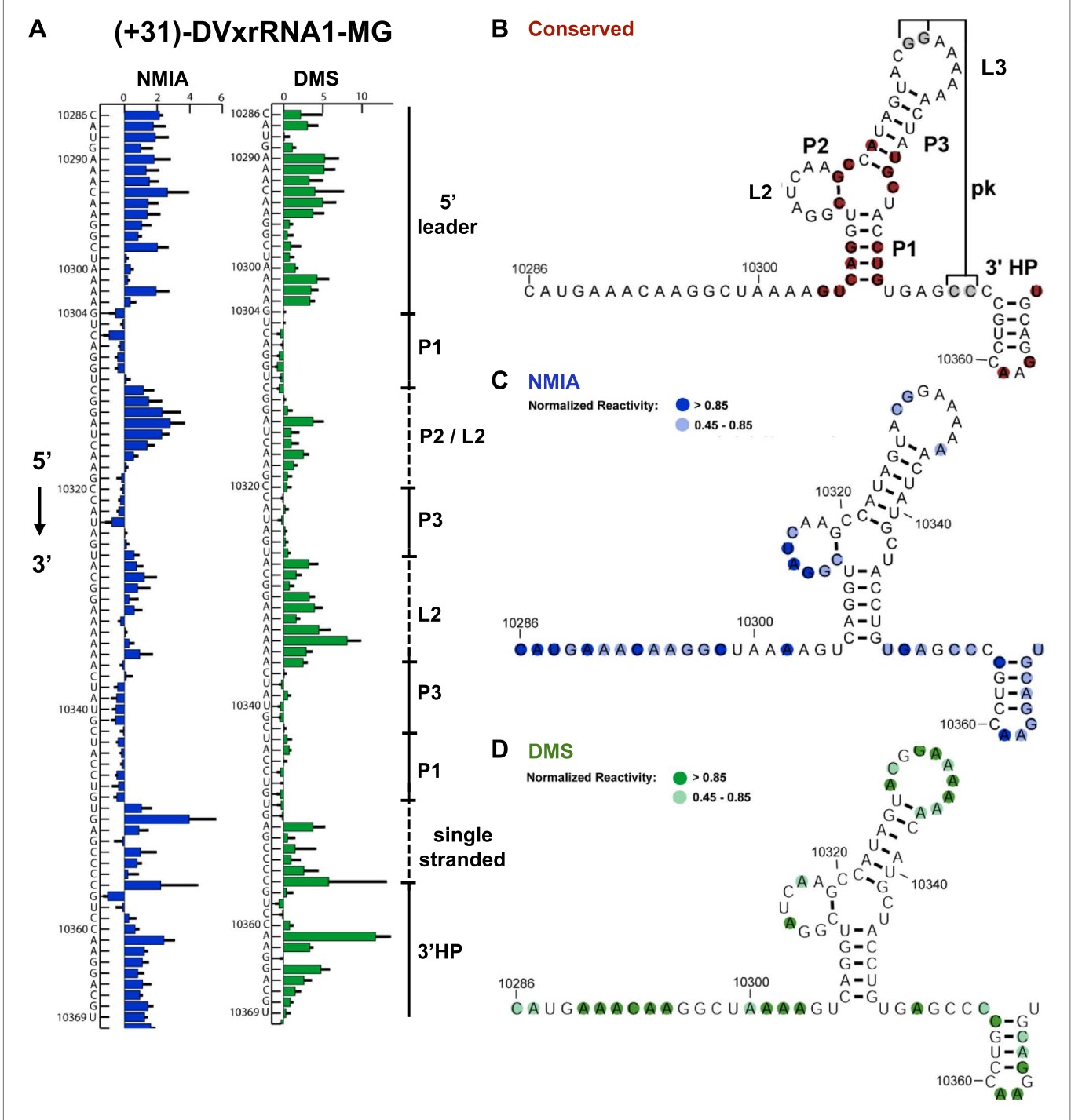

**Figure 3**. Refined secondary structure of SLII RNA in the DENV2 3'UTR. (**A**) Chemical probing profiles of (+31)-DVxrRNA1-MG with NMIA (blue) and DMS (green). The y-axis denotes the RNAs sequence/position, the x-axis depicts chemical reactivity. Data represent the average of three independent experiments, error bars are one standard deviation from the mean. Secondary structure elements are indicated to the right of the graphs. (**B**–**D**) Overlay of phylogenetic (**B**), NMIA (**C**), and DMS (**D**) data that support the shown refined secondary structure. In panel (**B**) secondary structure elements are labeled according to standard convention. The RNA shown here was quantitatively resistant to Xrn1 (**Figures 5B and 6D,E**).

The following figure supplements are available for figure 3:

**Figure supplement 1**. Naming Strategy for Xrn1-Resistant RNAs.

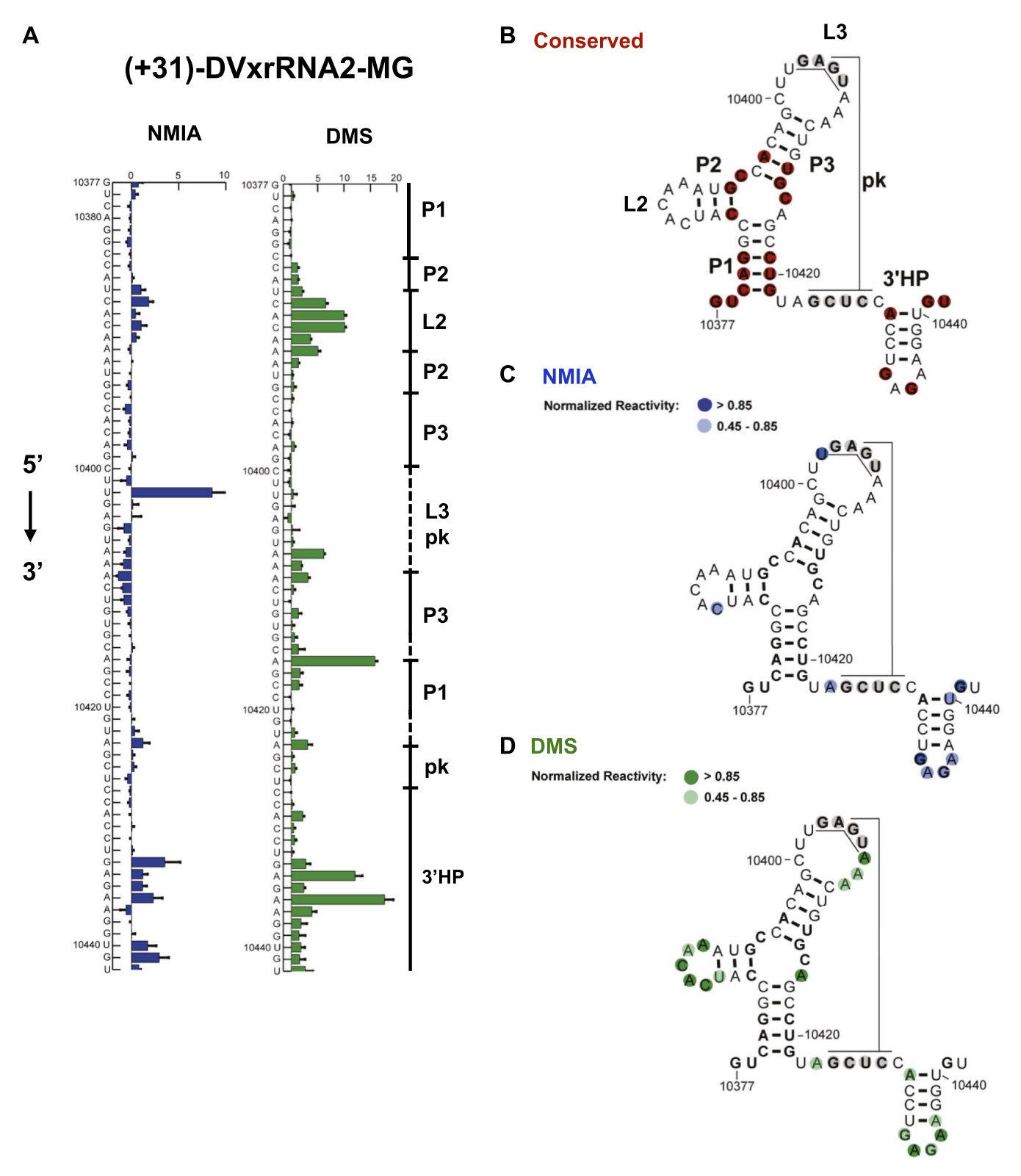

**Figure 4**. Refined secondary structure of SL IV RNA (DVxrRNA2) in the DENV2 3'UTR. (**A**) Chemical probing profiles of (+31)-DVxrRNA2-MG from NMIA (blue) and DMS (green) -probed RNAs. The y-axis contains the sequence and nucleotide numbers, the x-axis depicts chemical reactivity. Data represent the average of three independent experiments, error bars show one standard deviation from the mean. Secondary structure elements are indicated to

*Figure 4. Continued on next page*

*Figure 4. Continued*

the right of the graphs. (**B**–**D**) Overlay of phylogenetic (**B**), NMIA (**C**), and DMS (**D**) data the support this refined secondary structure. In panel (**B**) we assign names to the secondary structure elements according to standard convention. The RNA shown here was quantitative resistance to Xrn1 degradation (*Figure 6D,E*).

(*Figure 5A*; *Deana et al., 2008*; *Messing et al., 2009*), and render them effective substrates for Xrn1. In the same reaction these RNAs were treated with Xrn1 from *Kluveromyces lactis* (KlXrn1) (*Chang et al., 2011*) to degrade the RNA (*Figure 5B*). Truncation of the input RNA to a specific shorter product was interpreted as diagnostic of Xrn1 resistance.

Using this assay we first tested the SL II element of the DENV2 3′UTR for resistance to Xrn1 using a 5′ leader consisting of the 31 nucleotides that naturally precede SL II in the viral genome. Addition of BdRppH and KlXrn1 to a reaction containing this test RNA as well as a synthetically generated 5′-monophosphorylated 24 nucleotide-long internal control RNA (24-mer) resulted in complete degradation of the control RNA and truncation of the test RNA to a discrete product, indicative of Xrn1 resistance (*Figure 5B*). Because the enzymes used in this experiment were pure (*Figure 5—figure supplement 1*), this experiment establishes that Xrn1 resistance is conferred entirely by the SL II test RNA (no auxiliary proteins are needed). We therefore conclude that RNA elements from within the DENV 3′UTR as short as ~60 nucleotides can resist Xrn1 and that these RNAs operate effectively outside the context of the viral 3′UTR. Interestingly, we observed no enzymatic resistance using other commercially available exonucleases (*Figure 5—figure supplement 2*), suggesting this RNA is specifically resistant to Xrn1. For clarity, here we refer to this (and other) discrete RNAs using the functionally-descriptive name of 'Xrn1-resistant RNAs' (xrRNAs) in order to distinguish them from the longer sfRNAs. To facilitate discussion of xrRNAs in this and future work we propose a simple naming scheme for xrRNAs that is outlined in the (*Figure 3—figure supplement 1*). Using this nomenclature, we here refer to the first Xrn1-resistant structure in the DENV 3′ UTR as 'DVxrRNA1'.

## Testing each discrete structure for Xrn1 resistance

To augment the aforementioned assay and enable monitoring of Xrn1 resistance in a quantitative and time-resolved manner, we engineered a fluorescence-based assay based on the malachite green (MG) aptamer (*Grate and Wilson, 1999*; *Baugh et al., 2000*). The fluorescence of the triphenylmethylene MG dye is quenched when free in solution but dramatically increases when bound to the aptamer. This feature provides a fluorescent readout of the MG aptamer's integrity (*Figure 6A*; *Babendure et al., 2003*). We designed RNA substrates with the 40 nt-long MG aptamer placed downstream of each RNA structure to be tested for Xrn1 resistance (following a 4 nucleotide-long poly(U) linker) (*Figure 6B*). In solution with MG, these RNAs produce an ~2300-fold increase in MG fluorescence over free dye (*Babendure et al., 2003*). If addition of RppH and Xrn1 results in degradation of the aptamer-tagged RNA (no Xrn1 resistance), fluorescence disappears over time (*Figure 6C*, bottom). If the RNA being tested provides Xrn1 resistance, the MG aptamer remains intact and fluorescence persists (*Figure 6C*, top). A fuller description of the development and testing of this technique will be reported elsewhere (Chapman et al., in preparation). We used this fluorescence assay to test the Xrn1 resistance properties of the five discrete structures located within the DENV2 3′UTR (*Figure 6D*). The 5′ leader of each RNA consisted of nucleotides 10,273–10,304 of the DENV 3′UTR, which is degraded when present before DVxrRNA1 (*Figure 5B*) and each had the MG aptamer downstream. For DVxrRNA1, the test RNA was thus named (+31)-DVxrRNA1-MG and other input RNAs were named following the same convention (*Figure 3—figure supplement 1*). As expected, upon the addition of both enzymes (+31)-DVxrRNA1-MG maintains high fluorescence throughout the course of the experiment, consistent with Xrn1 resistance (*Figure 6D*). Likewise, SL IV demonstrated resistance to Xrn1-mediated decay (DVxrRNA2). In contrast, reactions with the DB2 and the 3′ terminal stem loop [test RNAs: (+31)-DVDB2-MG and (+31)-DV3SL-MG] show a decrease in fluorescence that is consistent with no Xrn1 resistance. The construct encoding DB1 [(+31)-DB1-MG] shows more intermediate behavior.

To assess the products of putative Xrn1 resistance from the fluorescence assay, we analyzed the reactions from the completed experiment using denaturing polyacrylamide gel electrophoresis (dPAGE) (*Figure 6E*). Consistent with the fluorescence traces, truncated Xrn1-resistant products accumulated in reactions with DVxrRNA1 and DVxrRNA2 (SL II and SL IV)-containing test RNAs. 3′ SL and

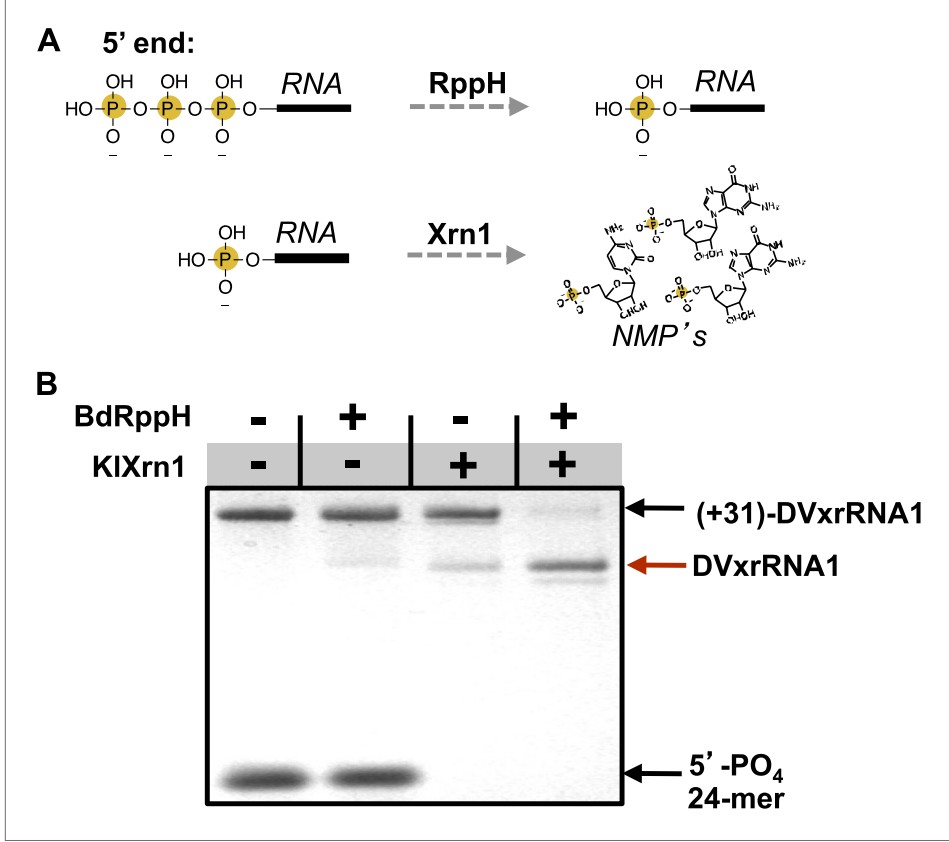

**Figure 5**. Recapitulation of Xrn1 resistance in vitro. (**A**) Reactions carried out by RppH and Xrn1. (**B**) Gel demonstrating the specificity of each component in the reconstituted reaction and the Xrn1-resistant behavior of (+31)-DVxrRNA1 (the input test RNA), which contains the isolated SL II element after a 31 nucleotide-long leader. The red arrow indicates the truncated product formed by Xrn1 resistance (DVxrRNA1, the resistant RNA). The 24-mer control RNA is labeled.

The following figure supplements are available for figure 5:

**Figure supplement 1**. Purification of RNA processing enzymes.

**Figure supplement 2**. DVxrRNA1 demonstrates specific resistance to Xrn1.

**Figure supplement 3**. Examination of *in trans* protection of other Xrn1 substrates by DVxrRNA1.

DB2-contianing constructs show complete degradation of the input RNA (*Figure 6E*), consistent with the fluorescence assay. In the case of the DB1-containing test RNA, the full-length RNA is partially depleted however there is no truncated product as would be expected if the DB1 sequence itself were Xrn1-resistant. The persistence of full-length RNA in this reaction suggests that Xrn1 may not be able to efficiently load onto this RNA and the lower intensity of the band suggests that when Xrn1 does load it degrades the RNA entirely. As >3 nucleotides of single-stranded RNA are required for Xrn1 to efficiently load on an RNA substrate (*Jinek et al., 2011*), the behavior of the DB1-containing RNA may indicate an alternate secondary structure that precludes the 5′ end of this RNA from being recognized by Xrn1. Cumulatively, our fluorescence and gel-based assays confirm two xrRNA structures in the DENV2 3′UTR that offer nearly quantitative protection to downstream RNA, of these we chose DVxrRNA1 as an archetype for deeper analyses.

## Mapping the location and nature of the product's 5′ end

We first mapped the precise location where Xrn1 is blocked by DVxrRNA1. To do this, we carried out a large-scale reaction using (+31)-DVxrRNA1-MG and recovered the truncated product RNA by dPAGE

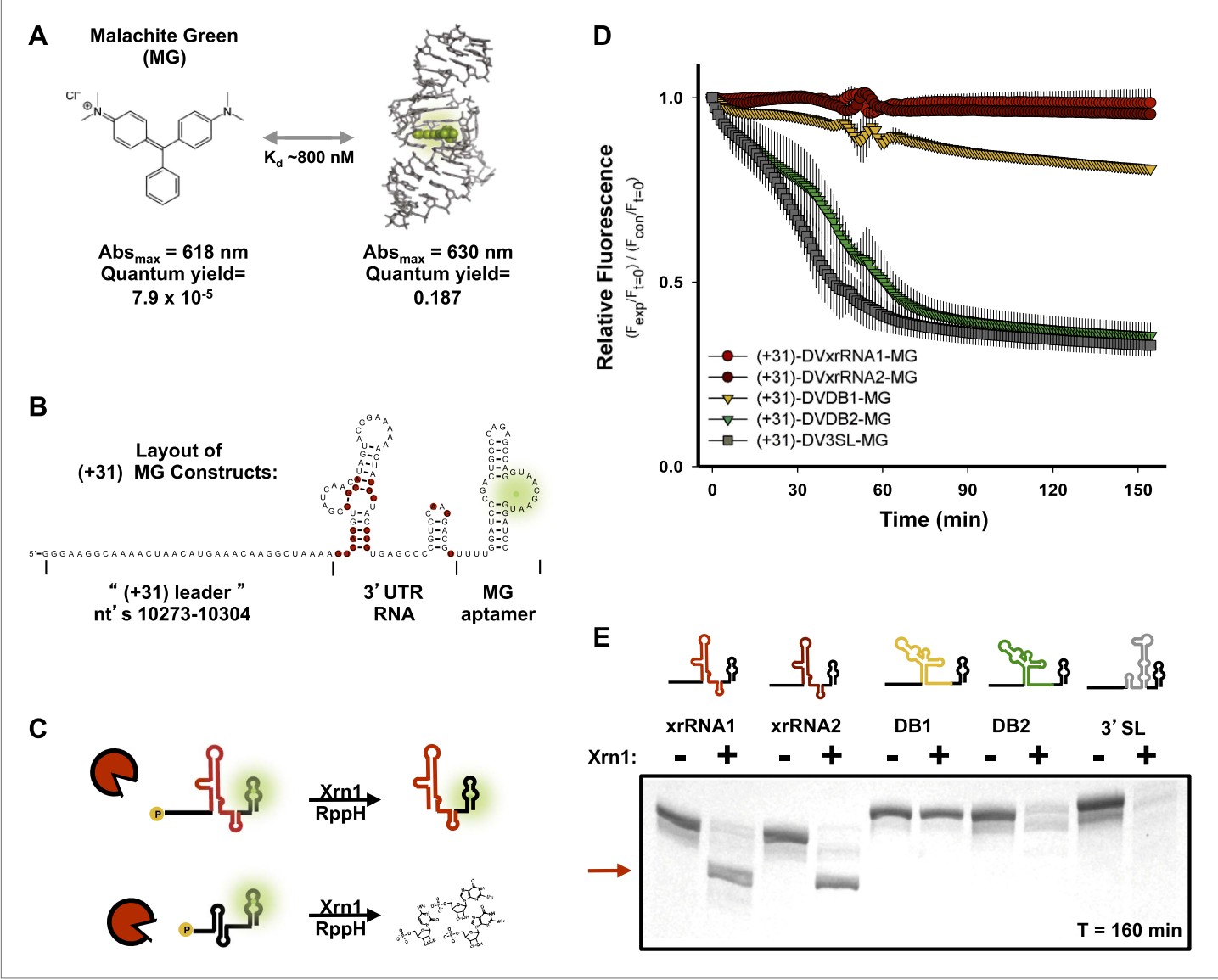

**Figure 6**. Design of a fluorescence assay for monitoring Xrn1 resistance and testing of individual DENV2 3'UTR structural elements. (**A**) Fluorescent properties and structure of the MG dye and MG aptamer. (**B**) Schematic of the RNA constructs used for monitoring the decay kinetics of individual elements of the DENV 3'UTR. Conserved nucleotides are colored red as in **Figure 3B**. Green indicates fluorescing MG dye. (**C**) Cartoon representing the expected outcome when using Xrn1-resistant (top) or non-resistant (bottom) RNAs. Green glow indicates fluorescence. (**D**) Fluorescence traces of different RNAs over the course of their reaction with Xrn1. X-axis is time and y-axis is relative fluorescence intensity. Data are normalized to (−) Xrn1 controls and are averaged over three independent experiments. Error bars show one standard deviation from the mean. (**E**) dPAGE analysis of the products of the experiment of panel (**D**).

(DVxrRNA1-MG$_{prod}$) (**Figure 7A**). We then used reverse transcription to determine the sequence of the 5'-end left by Xrn1 resistance (**Figure 7B**). This experiment reveals that DVxrRNA1-MG$_{prod}$ RNA ends at a U corresponding to U10299 of the DENV2 3'UTR, five nucleotides upstream of the first phylogenetically-conserved nucleotide in the DVxrRNA1 structure.

We then interrogated the chemical nature of the 5' end left by Xrn1, which speaks to a possible mechanism for Xrn1 resistance. Substrate recognition by Xrn1 requires a 5' monophosphate at the end of an RNA and each catalytic cycle regenerates this chemical species. Could Xrn1 resistance be based in part upon malfunction of this process? If so, the products formed by xrRNAs might not bear a 5' monophosphate and would be precluded from further degradation. We determined the

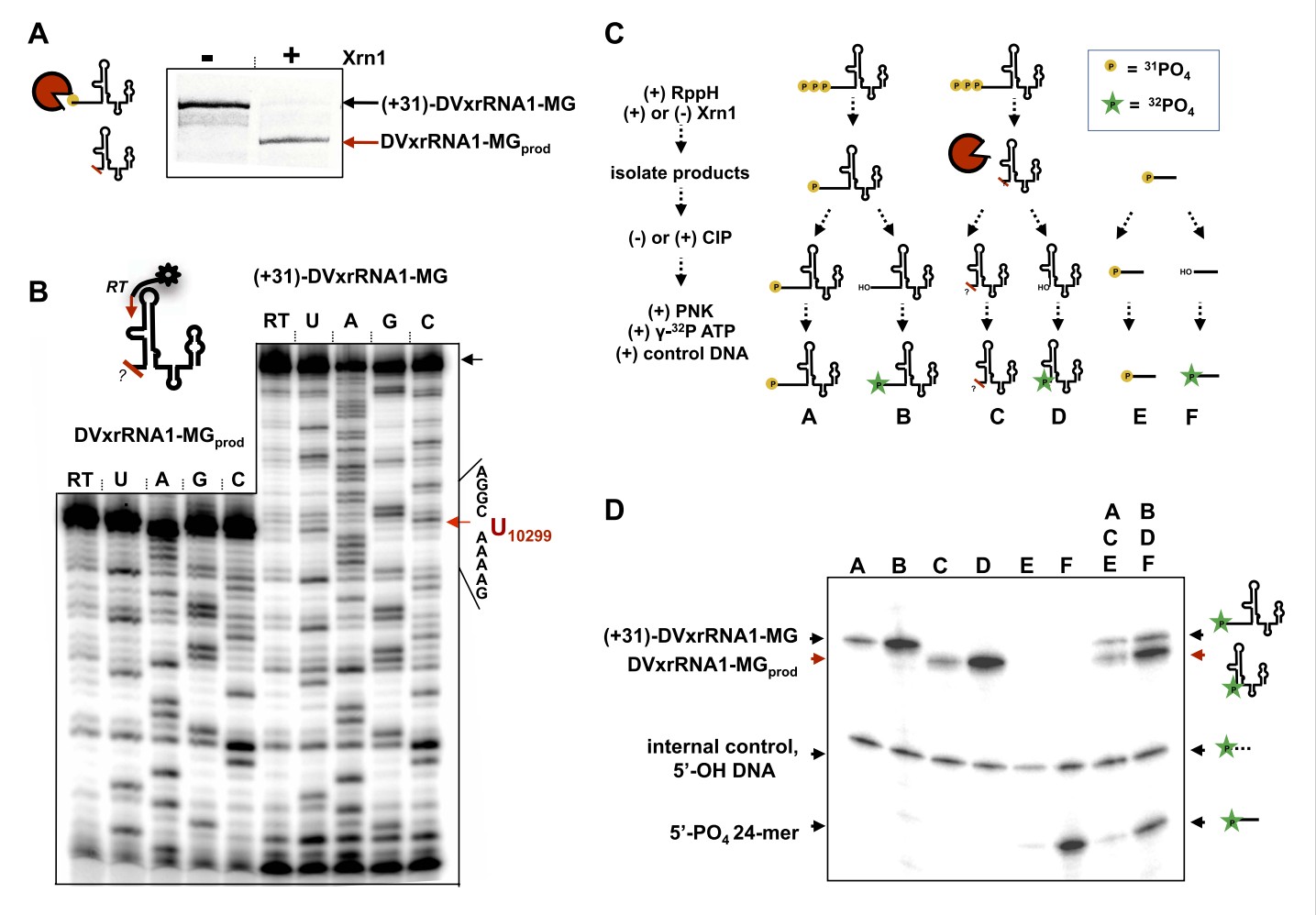

**Figure 7**. Characterization of the products left by Xrn1 resistance. (**A**) Gel of RNAs used to map the 5′ end of Xrn1-resistant product RNA. (**B**) Reverse transcription of RNAs from panel (**A**). The cartoon inset shows a schematic of the reaction. Dideoxy sequencing lanes are labeled. The location of the stop site (and hence the 5′ border of the product RNA) is shown with a red arrow to the right, along with the sequence of the RNA surrounding this position. (**C**) Diagram of the experiment used to determine the phosphorylation state of the products left by Xrn1 resistance. Yellow balls indicate non-radioactive phosphates, a green star depicts $^{32}$P radioactive phosphates. (**D**) Gel containing the outcome of the experiment shown in panel (**C**). Different species of RNA or DNA are labeled to the left and cartooned to the right. Red arrows: Xrn1-resistant product RNA.

phosphorylation state of the 5′ end of Xrn1-resistant products formed by DVxrRNA1 using the procedure outlined in *Figure 7C* and described in the Supplementary Description of 'Materials and methods'. Briefly, after 'product' and 'control' RNAs were isolated, half of each sample was dephosphoylated using calf intestinal alkaline phosphotase ('CIP'ed'). We then compared the ability of T4 polynucleotide kinase (PNK) to ligate a radioactive phosphate to the CIP'ed vs un-CIP'ed RNAs, expecting that the presence of a 5′-monophosphate would interfere with effective phosphorylation. In these experiments we included both a synthetically 5′ monophosphorylated 24-mer RNA and a 5′ hydroxylated 40-mer DNA as controls. As predicted, the dephosphorylated products of the DVxrRNA1 'control' RNA are better substrates for phosphorylation than their un-CIP'ed counterparts (*Figure 7D*, lanes A and B). Control reactions involving the synthetic 5′-monophosphorylated 24-mer produce similar results (*Figure 7D*, lanes E and F). Similarly, the CIP'ed Xrn1-resistant product RNAs are better substrates for phosphorylation than their un-CIP'ed counterparts. These results indicate that the RNA product of Xrn1 resistance are 5′-monophosphorylated and are still viable substrates for Xrn1 (*Figure 7D*, lanes C and D). Xrn1 resistance is thus not the result of chemical 'mis-step' of the exonuclease.

## Linking conserved sequence and structure and function

The results presented above and those in the literature lead to the hypothesis that Xrn1 resistance depends on a specific RNA structure. To link RNA sequence, structure, and function and to identify the elements of the DVxrRNA1 that are important for conferring Xrn1 resistance, we designed eleven mutants targeting conserved xrRNA sequence and secondary structure elements and tested their Xrn1-resistant properties (*Figure 8*). As before, we transcribed these RNAs with 31 nucleotide-long 5' leaders and compared their Xrn1-resistant behavior (relative to a wild-type (WT) control) using both dPAGE and our fluorescence assay. In these experiments DVxrRNA1 mutant RNAs displayed a range of Xrn1-resistant behaviors (*Figure 8A,B*):

### Importance of the PK

Mutants 1–3 tested the importance of the putative DVxrRNA1 PK. Mutants 1 and 2 cannot form the hypothesized PK and are less resistant to the Xrn1 than WT, but to only a moderate extent (64% and 88% resistance of WT, respectively). Mutant 3, in which the potential for a PK is restored, has an activity at 90% of WT. The observation that mutants 1 and 2 retain robust Xrn1 resistance is consistent with our chemical probing data that argue against the formation of a stable PK in the resistant RNA (*Figure 3C,D*). Overall, our data suggest a PK interaction may enhance Xrn1 resistance by DVxrRNA1 but is not essential to its operation and therefore is not the critical structural feature of this RNA.

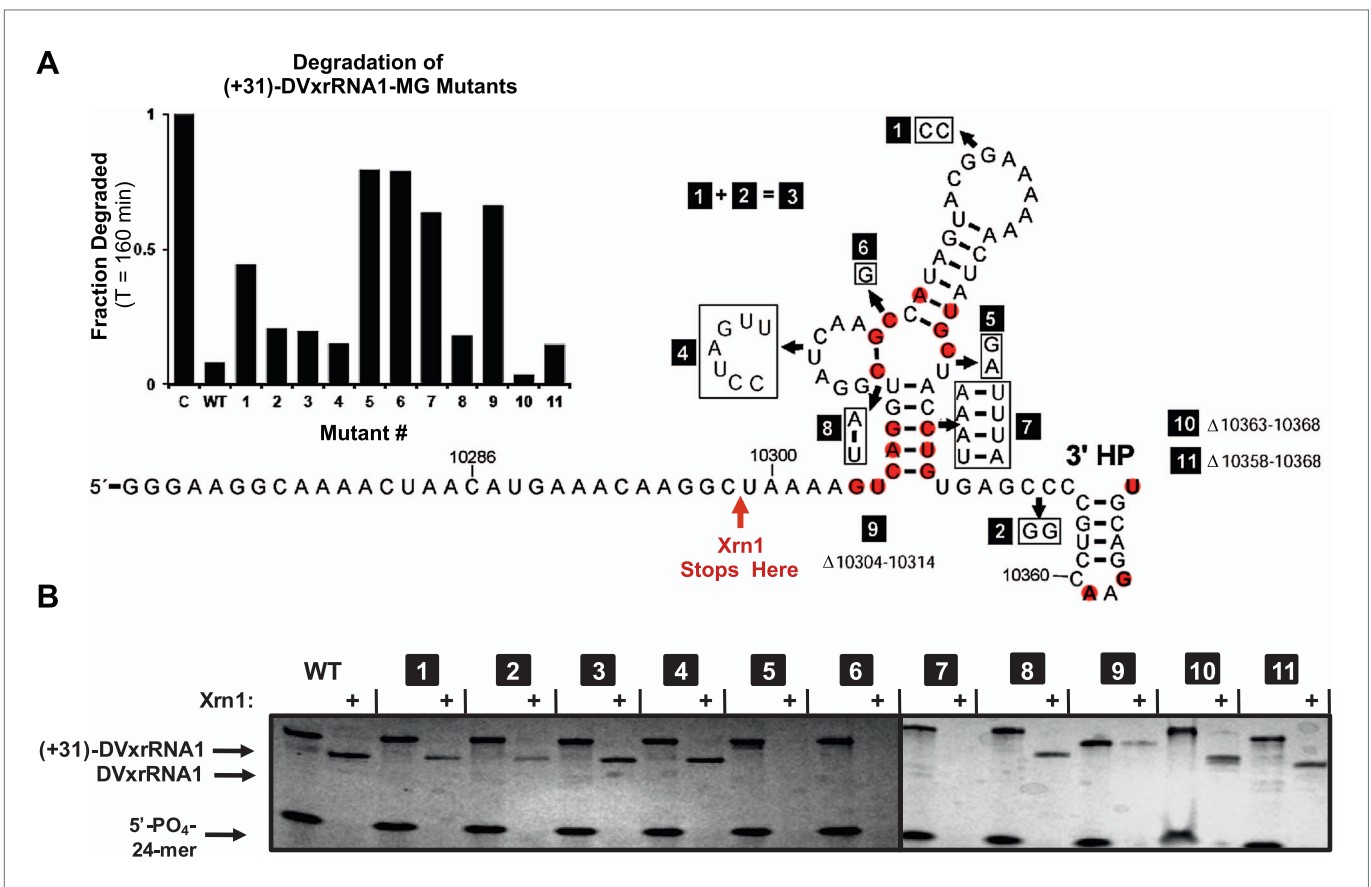

**Figure 8**. Mutational analysis of DVxrRNA1. (**A**) Secondary structure of DVxrRNA1 with mutations labeled Conserved sequence elements in red. The stop site for Xrn1 is indicated. Inset is a bar graph containing the effects of mutations Xrn1 resistance when quantified using the fluorescence assay. The x-axis identifies each mutant, the y-axis the fraction of MG-tagged RNA degraded at 160 min. C is a non-resistant control RNA. Graph depicts the average of two independent experiments. (**B**) Gel analysis of the reaction of each mutant with Xrn1 and RppH as in *Figure 5*.

The following figure supplements are available for figure 8:

**Figure supplement 1**. Native PAGE analysis of DVxrRNA1 mutants.

## Importance of L2

Mutant 4 tested the contribution of the L2 loop to Xrn1 resistance. When the 7 nucleotides in this region were mutated to their Watson-Crick (W-C) complement the RNA remained essentially fully resistant to Xrn1. This finding is consistent with the lack of phylogenetic conservation observed in this region.

## Importance of the three-way junction

Mutants 5 and 6 tested the importance of conserved nucleotides located within the P1-P2-P3 three-helix junction of DVxrRNA1. Both of these mutations essentially abolish resistance to Xrn1. These findings are consistent with the high conservation of these sequence elements. Furthermore, the dramatic loss of function in these mutants indicates the three-way junction is a critical element in conferring Xrn1 resistance and is likely the central element around which the structure organizes.

## Importance of paired bases

Mutants 7 and 8 tested the importance of the identities of the base pairs present in the P1 and P2 helices of DVxrRNA1. The motivation for exploring these mutations was based on the conservation of the sequences in these helical regions. In mutant 7, mutation of the natural G-C rich stem of P1 into a series of five A-U base pairs abrogates Xrn1 resistance, consistent with the need for a P1 helix of defined sequence. In contrast Xrn1 resistance is maintained in mutant 8, in which the C-G base at the base of P2 is exchanged for a U-A base pair. This result suggests accurate assignment of secondary structure in this region and reveals tolerance to mutations in this conserved base pair.

## Importance of the 5' and 3' ends

Mutants 9–11 tested the consequences of truncating the DVxrRNA1 structure from either the 5' or 3'' end. Deleting the first 10 nucleotides after the conserved 5' end of DVxrRNA1 (nucleotides 10,304–10,314, mutant 9) yielded an RNA that wasn't completely degraded over the course of the experiment but did not yield a truncated product indicative of Xrn1 resistance. This RNA likely has a 5' end structure that inhibits efficient Xrn1 loading. Mutations 10 and 11 tested truncations of the 3' end. Despite conservation of several nucleotides in this region, mutants 10 and 11 remain resistant to Xrn1. In conjunction with our experiments which define the 5' border of xrRNA1 we can therefore localize Xrn1 resistance to a region of the DENV2 genome as small as 59 nucleotides (nucleotides U10299–U10358).

## The three-way junction structure is essential for function

If Xrn1 resistance is due to the formation of a specific RNA structure, mutations that affect Xrn1 resistance should disrupt this structure. To test this, we subjected the mutants described above to non-denaturing (native) gel electrophoresis, which is sensitive to changes in the global structure of a folded RNA (*Figure 8—figure supplement 1*; *Woodson and Koculi, 2009*). In gels containing 5 mM $MgCl_2$, mutants of DVxrRNA1 display a spectrum of electrophoretic behaviors, indicating a variety of effects on RNA structure. One mutant that stood out was the non-Xrn1-resistant mutant 6 (C10320→G) (*Figure 9*). In EDTA-containing gels this mutant migrated at the same rate as WT DVxrRNA1, however in gels containing $MgCl_2$ mutant 6 demonstrates significantly retarded mobility, indicative of altered folding of the RNA. This suggests that a single point mutation made in the DVxrRNA1 three-helix junction is capable of disrupting a $Mg^{2+}$-dependent tertiary fold that is linked to Xrn1 resistance. To further examine this behavior we used chemical probing experiments to assess the structure of the mutant 6 RNA. The normalized NMIA and DMS reactivity profiles of the WT and mutant 6 MG aptamer-tagged RNAs are similar but with several marked differences (*Figure 9A,D*). These data reveal that the WT and mutant 6 RNA have similar secondary structures (consistent with the EDTA-containing native gels, *Figure 9C*) but indicate that the C10320→G mutation affects several structural elements and changes the global fold of the RNA (consistent with the $MgCl_2$-containing native gel *Figure 9C*). These findings support the conclusion that the DVxrRNA1 three-helix junction organizes a specific structure that is essential for Xrn1 resistance.

## xrRNA structural integrity is directly linked to sfRNA production during WNV infection

Having established that xrRNA function depends on a structure organized by the P1-P2-P3 three-helix junction, we then assessed the generality of our findings in another FV and tested their relevance to sfRNA production in infected cells. Because sfRNA formation is readily explored in the context of the Kunjin strain of WNV (WNV_KUN), we used this virus system to assess the generality and relevance. We

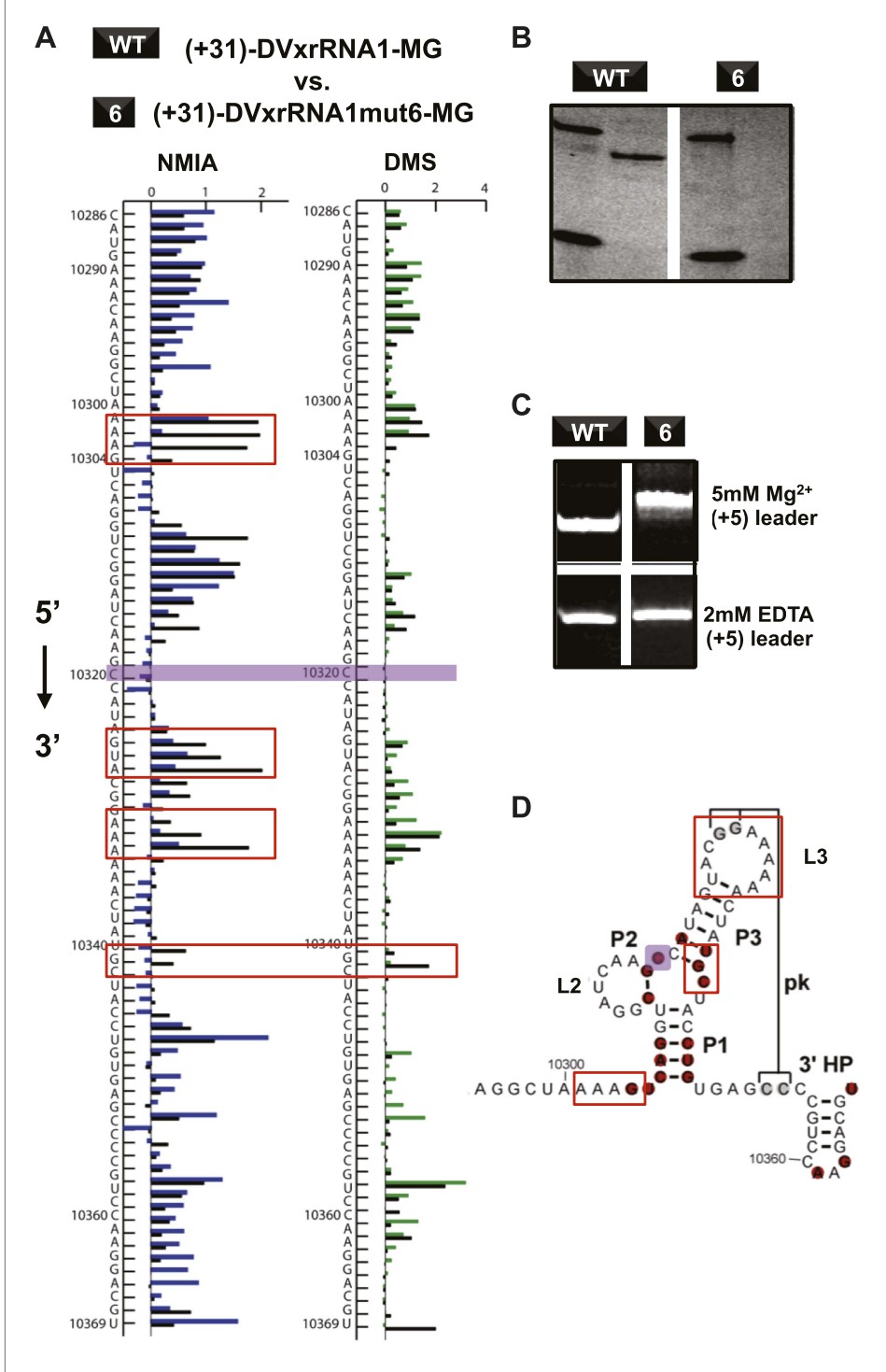

**Figure 9**. Structural analyses of DVxrRNA1 mutant 6 (mutant in the three-way junction). (**A**) Normalized NMIA and DMS reactivity profiles of mutant 6 compared to WT RNA, depicted as in **Figures 3 and 4**. Colored bars (green and blue) represent the WT RNA and black bars represent mutant 6. The location of the point mutation is shown with a purple shaded box. Reactivity changes are indicated by red boxes. (**B**) dPAGE of Xrn1 resistance by WT and mutant 6. (**C**) Non-denaturing (native) PAGE analyses of WT and mutant 6. (**D**) Secondary structure of the DVxrRNA1 with elements labeled. The purple shaded box shows the location of the mutation. Regions of the structure that show substantial changes in the chemical probing (from panel (**A**)) are indicated by red boxes.

first tested two proposed xrRNA structures located within in the WNV$_{KUN}$ 3'UTR (*Figure 10A*) for Xrn1 resistance, again using a 31 nucleotide-long leader. Both of these RNAs, WNV$_{KUN}$xrRNA1 (nucleotides 10,499–10,574) and WNV$_{KUN}$xrRNA2 (nucleotides 10,659–10,728), demonstrate resistance to Xrn1 in vitro (*Figure 10B*). We also tested versions of these RNAs containing C→G point mutations made in positions analogous to the mutation (mutant 6) that abolishes Xrn1 resistance in DVxrRNA1 (C10519→G in WNV$_{KUN}$xrRNA1 and C10680→G in WNV$_{KUN}$xrRNA2) (*Figure 10B,C*). We observe that the C→G point mutation in WNV$_{KUN}$xrRNA1 partially compromises Xrn1 resistance, while the same point mutation made in WNV$_{KUN}$xrRNA2 abolishes resistance. Together these results demonstrate the generality of the observations we made using DVxrRNA1 but also show some subtle variations in different xrRNA species.

The above results allowed us to examine the role played by each WNV$_{KUN}$ xrRNA structure in the formation of sfRNAs during infection. We infected human 293T cells with both wild-type and mutant WNV$_{KUN}$ and assessed sfRNA accumulation at 48 hours post infection (h.p.i.). Northern blots of total RNA isolates show that infection by wild-type WNV produces three prominent sfRNA species (*Figure 11*). Based on their size, the largest of these RNAs, sfRNA1 and sfRNA2, are the result of Xrn1 pausing at WNV$_{KUN}$xrRNA1 and WNV$_{KUN}$xrRNA2, respectively. The third species, sfRNA3 corresponds with previous observations of Xrn1 stopping prior to DB1 of WNV$_{KUN}$. Infections using a WNV$_{KUN}$ harboring a C10519→G point mutation in WNV$_{KUN}$xrRNA1 results in a significant and reproducible decrease in sfRNA1 formation, corresponding with the incomplete loss of Xrn1 resistance we observe in vitro. Likewise, infection with WNV$_{KUN}$ containing a C10680→G point mutation in WNV$_{KUN}$xrRNA2 results in the disappearance of sfRNA2, and again mirrors our findings in vitro. Finally, infection by a WNV$_{KUN}$ containing both C10519→G and C10680→G point mutations produces low levels of sfRNA1 and essentially no sfRNA2. The more substantial decrease in sfRNA1 formation in the double mutant as opposed to the C10519→G single mutant may suggest a degree of cooperation between the two xrRNA structures although establishing such an interaction will require additional experimentation. Cumulatively our results show that the structural and functional features we observe have applicability across xrRNAs from diverse FV's and that the structural characteristics that confer Xrn1 resistance in vitro

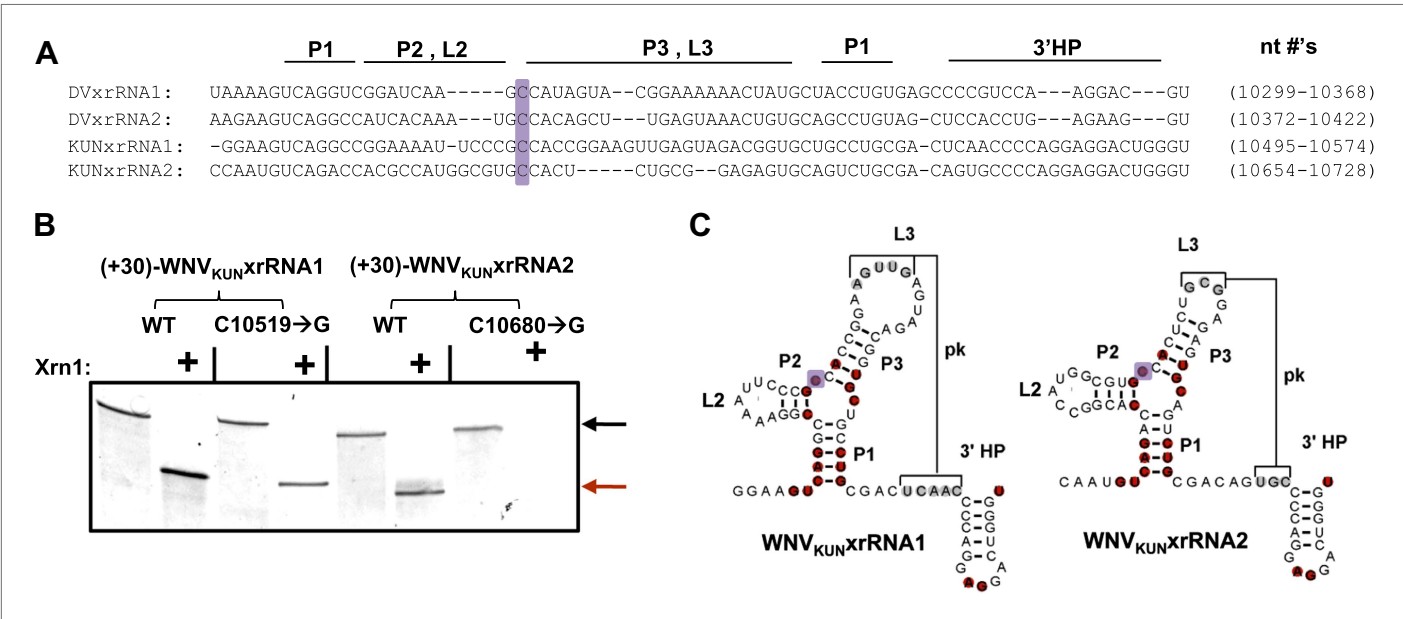

**Figure 10**. Identification and testing of WNV$_{KUN}$ xrRNA structures. (**A**) Sequence alignment of DVxrRNA1, DVxrRNA2, WNV$_{KUN}$xrRNA1 and WNV$_{KUN}$xrRNA2. Nucleotide positions are indicated and correspond to Genbank accession numbers M20558.1 and AY274504.1 for DENV2 and WNV$_{KUN}$, respectively. Conserved nucleotides of these RNAs are highlighted in red. The point mutation we made in the three-was junction is indicated with a purple box. (**B**) dPAGE of the Xrn1 resistance assay run using WNV$_{KUN}$ xrRNAs and C10519→G and C10680→G point mutants. (**C**) Conserved secondary structure of WNV$_{KUN}$xrRNA1 and WNV$_{KUN}$xrRNA2, with conserved nucleotides highlighted and the location of the mutation shaded purple.

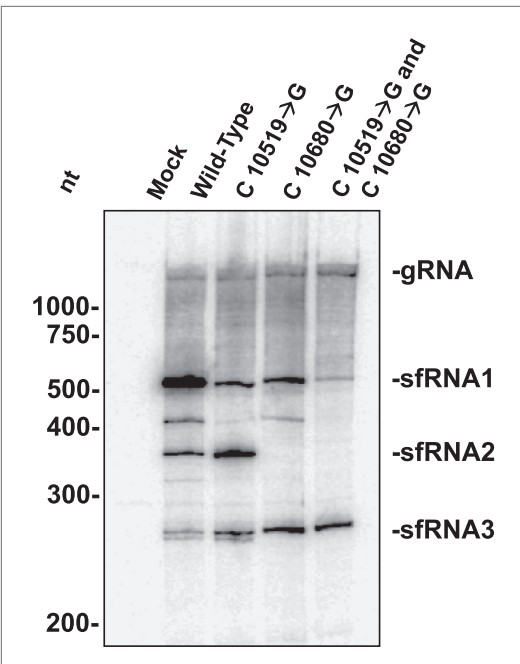

**Figure 11**. sfRNAs produced during infection by wild-type and mutated WNV_KUN. Northern blot analysis of total RNA isolated from human 293T cells infected with WT and mutants WNV_KUN, analyzed at 48 h.p.i. The location of molecular weight markers and the identity of sfRNA species are indicated to the left and right of the blot respectively.

corresponds directly to the accumulation of disease-causing sfRNAs during viral infection.

## Discussion

Non-coding RNAs play important roles in viral disease (*Steitz et al., 2011*). During infections caused by FV's, the formation of sfRNAs through incomplete degradation of the viral genome is directly linked to disease (*Pijlman et al., 2008*; *Liu et al., 2014*). In this study we identified discrete elements of the DENV2 3'UTR that confer resistance to Xrn1 (xrRNAs) and explored characteristics of the RNA structures responsible. We correlated these findings with similar RNAs in WNV_KUN and demonstrated that the formation of specific, discrete folded structure is responsible for the production of sfRNAs. Here we discuss implications of these findings as they relate to the organization of the FV 3'UTR and the infection strategy used by these viruses. Based on our data we propose a model for how an RNA structure might confer Xrn1 resistance, and speculate about implications for other Xrn1-resistant RNAs.

The DENV 3'UTR (and correspondingly its sfRNA) is comprised of individual structural elements that, based on our probing data, do not appear to interact with each other to form a higher-order structure. This differs from other viral 3'UTRs that we have shown can fold into a single higher-order structure (*Hammond et al., 2010*). The presence of individually folded, discrete structures within the DENV 3'UTR correlates well with the observation that different elements are responsible for interacting with different proteins and perform different tasks during the FV replication cycle (*Polacek et al., 2009*; *Manzano et al., 2011*; *Ward et al., 2011*; *Hussain et al., 2012*). This 'knots in a rope' architecture of the DENV2 3'UTR and sfRNA may be important to enable and organize these multiple functions.

We identified two resistant functional RNA structures (DVxrRNA1, DVxrRNA2) located in succession near the 5' end of the DENV 3'UTR. These elements contain previously-proposed secondary structure stem-loop elements SL II and SL IV. We did not observe resistance from the DB elements in vitro; if they confer Xrn1 resistance during infection this may require interaction with a protein factor. The position of the two confirmed xrRNAs is consistent with a role in protecting downstream sequences and structures, including the DBs and 3'SL from degradation. Many other FV's also contain two neighboring xrRNA structures at the 5' end of their 3'UTR's (a notable exception is YF, which only contains one) (*Pijlman et al., 2008*). The evolutionary conservation of this pattern suggests that downstream RNA elements are critical for sfRNA function and viral success; however, the nearly quantitative resistance to Xrn1 demonstrated by these structures begs the question: why there are two xrRNAs in series when seemingly one should suffice? Perhaps two xrRNAs provide redundancy to ensure production of the sfRNA or perhaps each xrRNA is tuned to operate most efficiently under different intracellular conditions encountered during the infection cycle (*Villordo and Gamarnik, 2013*). Our data suggest that the PK interaction is stable in only one of the two xrRNAs in DENV2, hinting that subtle functional differences may exist between the two structures. Furthermore, our data from cells infected with mutant Kunjin Virus hint at cooperation or coupling between these elements.

Cumulatively, our data suggest that a critical feature of Xrn1 resistance is the formation of a specifically structured three-helix junction. The presence of multiple conserved bases within the three-way junction, including several unpaired nucleotides, suggests that this core organizes the tertiary fold of the RNA. This idea is supported by that fact that this fold can be disrupted by a point mutation made

in one of the unpaired nucleotides within the junction. The fold this junction organizes is important for Xrn1 resistance, and while it may be further stabilized by a PK, this tertiary interaction does not appear to be necessary for Xrn1 resistance in vitro (at least in the context of DVxrRNA1).

Full understanding of the basis of Xrn1 resistance will require a high-resolution structure of an xrRNA, but our results allow some predictions. First, examination of the three-way junction features and comparisons with other RNA junctions suggest that it forms a 'type-C' three-helix junction in which P1 and P2 coaxially stack and P3 assumes an acute angle relative to P1 (*Lescoute and Westhof, 2006*). Based on this, we predict that folding of the junction (in the absence of any tertiary interactions) would bring L3 close to the base of P1. Formation of the PK would further stabilize this helical arrangement. How does this conformation result in Xrn1 resistance? We speculate that as the 5' end of the RNA is drawn into the active site, the enzyme encounters a structure formed by the spatially adjacent P1 and P3 helices that prevents the helicase activity of the enzyme from unwinding the RNA. *Jinek et al. (2011)* have proposed that RNA entering the active site of Xrn1 is unwound when pulled through a narrow channel formed by the α1 helix and an adjacent loop. The xrRNA may have evolved an unusual and specific structure that prevents these features of the enzyme from effectively interacting with and unwinding the RNA. Such a structure need not be rigid or even exceptionally thermodynamically stable, but must be arranged in such a way that the enzyme cannot process it.

We did not directly address the pathways by which viral genomic RNAs could be decapped and recognized by Xrn1, or the pathways by which sfRNA alters infected cells once it has been formed, but our results may lend insight into these processes. One interesting idea is that sfRNA production causes dysregulation of Xrn1-dependent mRNA turnover in the infected cell. Indeed, studies of Dengue and Kunjin virus infections showed that sfRNA production stabilizes mRNAs by inhibiting Xrn1 (*Moon et al., 2012*). One possible explanation of this effect could be that Xrn1 remains bound to xrRNAs, perhaps locking the enzyme in an inactive conformation or sequestering it from other substrates. There is evidence that an RNase L-inhibiting RNA of poliovirus operates as a competitive inhibitor through this type of mechanism (*Keel et al., 2012*). However, in several of our experiments Xrn1 processes over 150 copies of the xrRNA substrate and over 300 copies of the 24-mer control RNA, indicating multiple turnover by the enzyme. Nonetheless, we did find conditions under which afforded some protection to an Xrn1 substrates *in trans* (*Figure 5—figure supplement 3*), but we have been unable to obtain evidence of formation of a stable complex between a resistant RNA and Xrn1 in a purified system (data not shown). Overall, these observations argue against a simple binding model for sfRNA-induced changes in mRNA turnover and suggest that sfRNA may act as a reversible inhibitor of Xrn1, perhaps changing the kinetics of the enzyme in infected cells.

That Xrn1 resistance is conferred by a discrete, relatively short, and highly conserved RNA structure raises the notion that it could be targeted with therapeutics. The observation that mutation of a single conserved nucleotide in the three-way junction completely abrogates Xrn1 resistance and sfRNA formation in human cells suggests that only a few key intermolecular interactions need to be altered to prevent xrRNA folding and function. Perhaps mutations or deletions in individual xrRNA elements present in other FV genomes could be become part of a general route toward the development of attenuated vaccines.

In conclusion, we mapped the architecture of the complete 3'UTR of DENV2 and identified Xrn1-resistant activity residing in discrete and portable structural elements. We identified similar Xrn1-resistant RNA structures in WNV$_{KUN}$ and demonstrated that these structures are directly linked to the formation of sfRNAs in vivo. Our studies regarding the characteristics of these uniquely, Xrn1-resistant RNAs suggest that a specific and unusual structure confers resistance to Xrn1. These studies provide a framework for more detailed structure-based mechanistic investigations of these RNAs that are directly linked to disease.

## Materials and methods

### General procedures

#### Chemical reagents and synthetic DNA

General chemical reagents were all of molecular biology grade or higher. Reagents in which high purity is believed to have been beneficial are indicated and their manufacturer's listed. All aqueous solutions were made using diethylpyrocarbonate (DEPC) treated milli-Q water and routinely filtered through 0.22 μm sterile filtration systems (Millipore, Billerica, MA). DNA primers were purchased from

Integrated DNA Technologies (IDT, Coralville, IA) and used without further purification. Nucleic acid concentrations were routinely determined by monitoring a solution's absorbance at 260 nm using a Nanodrop UV-Vis spectrophotometer.

## Plasmid/primer design

The sequence of a plasmid containing the 3' UTR of the Jamaica/N.1409 strain of a serotype 2 Dengue virus (GenBank accession number M20558.1) was verified by Big Dye sequencing at UC AMC core facilities. To make dsDNA templates for transcription, appropriate regions of this plasmid were amplified by PCR and transcribed as diagrammed in *Figure 2—figure supplement 1*. Alternatively, the (+31) 'leadered' mutants of DVxrRNA1 were made using DNA templates generated by PCR amplification of overlapping primers. The specific sequence of any primer is available by request from the authors.

## RNA synthesis

### PCR amplification of transcription templates

dsDNA templates were routinely made through two stages of PCR. First stage PCR was routinely carried out on the 50 µl scale. 20 µl of the first reaction was then used as a template for a second reaction carried out on a 1 ml scale. Typical PCR conditions: 25 ng plasmid DNA or 0.2–0.4 µM duplex DNA template, 0.5 µM forward and reverse primers, 250 µM dNTPs, 2 mM $MgSO_4$, 10 mM KCl, 10 mM $(NH_4)_2SO_4$, 20 mM Tris pH 7.9, 0.1% Triton X and Pfu DNA polymerase. PCR efficiency was verified by 1.5% agarose gel electrophoresis.

### Transcription

1 ml PCR reactions were typically used as a direct input into 5 ml transcriptions. Transcription reactions contained: ~0.1 µM template DNA, 8 mM NTP's, 60 mM $MgCl_2$, 30 mM Tris, pH 8.0, 10 mM DTT, 0.1% Spermidine, 0.1% Triton X, and T7 RNA polymerase produced within our lab. 1–3 µl of 0.8 U µl RNasin Plus RNase Inhibitor (Promega, Madison, WI) was occasionally included. Reactions were typically incubated 8–12 hr at 37°C.

### Transcription clean-up

Following transcription, insoluble inorganic pyrophosphate was removed by centrifugation. Reactions were brought to 2 M LiCl, stored at −20°C for 2–16 hr and then precipitated by centrifugation. Pelleted RNA was resuspended in 3 ml of 300 mM EDTA, 100 mM triethylammonium acetate (TEAA), pH 7.0 prior to HPLC purification.

### HPLC purification

RNAs were purified using an Agilent 1260 Infinity HPLC using gradients of 1–30% acetonitrile in 100 mM TEAA, pH 7.0 over a Varian PLRP-S 8 µm 150 × 7.5 mm column. RNA containing fractions were pooled, concentrated and finally buffer exchanged into water using Amicon Ultracel concentrators. RNA was then stored at −20°C until used.

## Protein synthesis

### KlXrn1 production

A plasmid encoding a 6XHis-tagged version of the previously crystallized form of Xrn1 from *Kleuveromyces lactics* (*Chang et al., 2011*) (residues 1–1245) was kindly given to us by Prof Dr Liang Tong at Columbia University. We expressed the recombinant protein in *Escherichia coli* and purified the enzyme according to the published procedure. The purity of KlXrn1 was verified by crystallization (data not shown) and SDS-PAGE (*Figure 5—figure supplement 1*). The activity of Xrn1 is typically defined as a weight of total organismal RNA digested per unit time. Based on the decay kinetics we observe using the fluorescence assay described in this work, we estimate the activity of our Xrn1 stocks to be roughly ~3 U/µl when a unit is described as amount of enzyme required to digest 1 µg of an ~150-mer 5'-monophosphorylated RNA (~80 pmol) in 1 hr at room temperature.

### BdRppH expression

A plasmid encoding the RppH from *Bdellovibrio bacteriovorus* (*Messing et al., 2009*) (BdRppH) was kindly provided to us by Dr Joel Belasco at the Skirball Institute, New York University School of Medicine. A6X His-tagged version of BdRppH was expressed in *E.Coli* and purified by $Ni^{2+}$-NTA affinity and size-exclusion chromatography. The purity of RppH produced was verified by SDS-PAGE

(*Figure 5—figure supplement 1*). We have not directly determined the activity of our RppH stocks although we know from control experiments that the enzyme is operating at least as fast as Xrn1. Therefore, when defined as the amount of enzyme required to convert 1 µg of an ~150-mer triphosphorylated RNA (~80 pmol) to an Xrn1-susceptible substrate in an 1 hr at room temperature, the activity of our RppH stocks was at least 3 U/µl.

Phosphodiesterase I from *Croatalus adamanteus* venom (3''→5', CaPde I) and Phosphodiesterase II from bovine spleen (5'→3', BtPdeII) were obtained as lyophilized solids (Sigma-Aldrich, St. Louis, MO). To create working stocks of each enzyme approximately ~5 mg of either white solid was dissolved in 500 µl of 50% glycerol, 200 mM NaCl, 50 mM Tris, pH 7.5. According to the manufacturer's specifications this results in an $\sim 8 \times 10^{-4}$ U/µl solution of CaPdeI and a 0.05 U/µl solution of BtPdeII. Reactions of (+30)-DVxrRNA1 with these enzymes were conducted as described for reactions involving Xrn1 except that they were incubated at 50°C. Reactions using CaPdeI were supplemented with 10 mM $Zn(OAc)_2$, which has been reported to be an essential cofactor of the enzyme.

During the course of these studies we also tested commercially available versions of Xrn1 and RppH (New England BioLabs, Ipswich, MA). These enzymes can be used to reproduce our findings.

## RNA folding

In general, RNA was refolded prior to each experiment described in this manuscript. This was done using a thermocycler program that first ramped to 90°C for 2 min, held at 20°C for 5 min, and finally cooled and held at 4°C until used.

## Standard buffer conditions

The majority of experiments described here were carried out in 100 mM NaCl, 10 mM $MgCl_2$, 50 mM Tris, pH 7.9, 1 mM DTT. For simplicity we will refer to these conditions as 1X EC3 throughout this supplement.

Denaturing polyacrylamide gel electrophoresis was carried out using gels consisting of 8M urea, 1X TBE (80 mM Tris, 80 mM borate, 2 mM EDTA, pH 8.0).

## Chemical probing strategy

In order to streamline experimentation and accommodate technical aspects of capillary electrophoresis, individual regions of the Dengue 3'UTR were transcribed between adjacent 5' and 3' 'cassettes'. These cassettes were uniformly installed before and after each of the five shorter RNA constructs as diagrammed *Figure 2—figure supplement 1*. The 5' cassette consisted of the first 31 nucleotides of the DENV2 3'UTR. This region of RNA is predicted to be predominantly unstructured in the context of these RNAs, agreeing well with the high level of modification we observe in our experiments. The 3' cassette consisted of a 4 nucleotide-long uridine tract followed by a malachite green aptamer. In addition to providing a convenient place to anneal a primer for reverse transcription, this fluorophore binding aptamer was later used to observe the Xrn1 decay kinetics of these RNAs.

## Probing reactions

In chemical probing experiments 2 µM RNA was folded in 1X EC3 and then reacted with NMIA or DMS. The RNAs mapped in this study were each subjected to a titration of these reagents in order to optimize the extent of modification and the corresponding efficiency of reverse transcription. When the full length 3'UTR was mapped using a primer corresponding to the 3' end of this RNA we observed significant background pausing in RT just prior to the 3' component of the DB1 pseudoknot (data not shown). We therefore designed a second, internal primer to monitor chemical modification 5' of this position in the full length 3'UTR/sfRNA (*Figure 2—figure supplement 1*).

- DMS modification was carried out in 10 µl reactions containing 2 µM RNA and 57 mM DMS in 1X EC3. Reactions were incubated for 5 min at 0°C then quenched by the addition of 1 µl of neat β-mercaptoethanol.
- NMIA modification was carried out in 10 µl reactions containing 2 µM RNA and 2.5 mM NMIA (Invitrogen) in 1X EC3. Reactions were incubated at 37°C for 1 hr.
- Following chemical modification, reactions were passed over homemade G-25 Sephadex (GE Healthcare, Little Chalfont, United Kingdom) spin columns and recovered. 1.25 pmol (0.1 µM, 0.063 equivalents) of an appropriate 5' 6-FAM end-labeled primer was then added. Reactions were then re-annealed in order to hybridize these primers to each RNA. Reverse transcription was then

carried out using Superscript III reverse transcriptase in a reaction containing 0.1 µM primer and ~1.5 µM RNA, 80 mM KCl, 3.23 mM MgCl$_2$, 6 mM DTT, 55 mM Tris, pH 8.3, and 533 µM dNTPs.

- Following reverse transcription, these reactions were again desalted using G-25 spin columns and brought to 50% formamide (HiDi, Applied Biosystems, Inc., Foster City, CA). 1 µl of a ROX-labeled DNA ladder (GeneScan-500, ABI) was then added to each ~40 µl reaction. Reactions were then heated at 90°C for 10 min in order to denature cDNA products from the RNA templates and subsequently transferred to 96-well plates prior to capillary electrophoresis.
- Capillary electrophoresis was carried out using an ABI 3500 Gene Scanner equipped with a 50 cm capillary unit filled with POP-7 polymer (ABI).

## Data processing

Capillary electrophoresis data were exported as .fsa files and processed using the HITRACE-web online software (*Kim et al., 2013*). This software was developed by and is maintained by the Das lab at Stanford University and Yoon lab at Seoul University in South Korea. The data files produced by HiTRACE were subsequently transferred to Matlab for further analysis, normalization, figure making and conversion into formats compatible with further processing (e.g., components of the RNAstructure suite) (*Reuter and Mathews, 2010*). Because the Dengue 3′UTR is predicted to harbor multiple pseudoknot structures, we used the recently developed software program Shapeknots (*Hajdin et al., 2013*) to incorporate our chemical mapping data into secondary structure prediction algorithms capable of identifying such structures. Figures overlaying chemical probing data onto RNA secondary structures were made using the software program R2R (*Weinberg and Breaker, 2011*).

## Xrn1 resistance experiment

### Standard Xrn1 resistance assay

A standard assay for determining Xrn1-resistance was developed in which 4 µg of RNA (typically ~100–200 pmol or ~2.5–4 µM RNA depending on the specific construct being evaluated) was folded in 36 µl of 1X EC3. In some experiments 300–450 pmol (~10 µM) of a synthetically prepared 5′-monophosphorylated 24-mer (IDT) was also included as an internal control for Xrn1 activity. Following refolding, 2 µl of >3 U/µl BdRppH was added and the reaction was split between two tubes. 1 µl of >3 U/µl Xrn1 was added to one half of the reaction while the other served as a (−)Xrn1 control. Reactions were incubated 6 hr at 37°C. Reactions were then quenched by the addition of an equal volume of a dPAGE loading dye containing 30 mM EDTA, 8 M Urea and 0.1% (wt/vol) xylene cyanol and bromophenol blue. RNA products were analyzed on 12–20% dPAGE gels and visualized by staining with methylene blue.

## Xrn1 decay kinetics monitored by the fluorescence of the malachite green aptamer

To assess RNA decay kinetics through monitoring the fluorescence of the MG aptamer, 100 µl reactions of 4 µM RNA (400 pmol, ~16 µg) were refolded in 1X EC3. Often several 100 µl reactions using the same RNA were carried out simultaneously and were pooled after being folded separately in an 8-tube PCR strip. While reactions were still pooled, 5 µl of >3 U/ µl RppH per reaction and 7.5–10 equivalents of malachite green (30 µM final concentration, 3 µmol per reaction) were added. 105 µl aliquots of this mixture were then divided into separate wells of a black, flat bottomed 96-well plate (Greiner Bio-One, Monroe, NC). Fluorescence was monitored using a Glomax Multi+ plate reader using a filter set that allowed excitation at 625 nm and monitoring of emission over 660–720 nm. Prior to the addition of Xrn1, fluorescence was measured for 5 min in order to ensure reactions had reached equilibrium and to allow RppH to start producing 5′-monophosphorylated Xrn1 substrates. At 5 min, 5 µl of >3 U/µl Xrn1 was added and the reaction was then monitored for the next 160 min. Under these conditions RNA decay rates vary linearly with added Xrn1 (data not shown). To produce the gel in *Figure 6E*, after the experiment had ended, 20 µl of each reaction was removed and subjected to 10% dPAGE. The gel was stained with methylene blue.

## Isolation of Xrn1-Resistant RNA products

In order to characterize the products left by Xrn1 resistance we conducted the experiment described for monitoring Xrn1 decay kinetics using (+31)-DVxrRNA1-MG RNA on an eightfold scale. The products of these reactions, along with an equivalent number of (−)Xrn1 control reactions, were recovered

after 160 min, pooled and subjected to 15% dPAGE. RNA products were visualized by staining with methylene blue, carefully excised and eluted from the gel overnight in water. Eluted RNA was concentrated using Amicon molecular weight cutoff filters and frozen at −20°C before additional analyses. Subsequent quantification revealed Xrn1-resistant RNA products were recovered with a 16% overall yield or as 0.52 µmol from a 3.2 µmol prep.

## Determination of the Xrn1 halt point on DVxrRNA1

Recovered DVxrRNA1-MG$_{prod}$ (product) and (+31)-xrRNA1-MG (control) RNAs were annealed to a 5′ end-labeled primer complementary to nucleotides 10,328 through 10,342 in DVxrRNA1. Reverse transcription was carried out as essentially as described for chemical mapping experiments. Reactions were quenched by the addition of 8 M urea, 10 mM EDTA loading dye and then heated to 90°C for 10 min in order to promote dissociation of cDNAs from the RNA. Reactions were then loaded onto 10% sequencing gels. Electrophoresis was carried out at 50 W for 4 hr. The resulting gels were removed, dried and exposed to Molecular Dynamics phosphor screens before being imaged on a Storm 720 scanner. Gels were visualized using ImageQuant software.

## Determination of the 5′ phosphorylation state of the Xrn1 resistant product

The workflow for this experiment is outlined in *Figure 7C*. To determine the phosphorylation state of Xrn1-resistant RNA products and accompanying controls, fractions of these samples were with first treated with calf-intestinal alkaline phosphotase (CIP) (line iii, *Figure 7C*). CIP reactions were carried out using ~5 µg of each RNA in a reaction performed as prescribed by the manufacturer of the enzyme (NEB). CIP'ed RNAs were purified by phenol chloroform extraction, washed once with 24:1 chloroform to isoamyl alcohol and then ethanol precipitated (70% ice cold EtOH, 300 mM NaOAC pH 5.2, −20°C, 2 hr). CIP'ed RNAs were then resuspended and quantified. All RNAs: CIP'ed and un-CIP'ed, 'product', 'control' and '24-mer' were then entered into phosphorylation reactions containing 50 pmol γ-$^{32}$P-ATP and 8 pmol of each RNA (line iv, *Figure 7C*).1 pmol of an synthetic, 5′-hydroxylated 40mer DNA was included to serve as a internal control for PNK activity. RNAs were then phosphorylated using T4 PNK (NEB) in the buffer supplied by the manufacturer for 1.5 hr at 37°C. Individual samples were diluted and then passed over homemade G-25 Sephadex columns to remove unincorporated γ-$^{32}$P-ATP. An equal volume of each reaction was the subjected to 15% dPAGE. The resulting gels were analyzed by phosphorimaging as described above.

## Native gel electrophoresis

Prior to native gel electrophoresis 2 µg of each DVxrRNA1 mutant was brought to 18 µl in water (~4 µM), refolded and then held at 4°C. Samples were brought to 1X EC3 by the addition of 2 µl of a 10X buffer and equilibrated on ice for 10 min. An equal volume of 50% (wt/vol) sucrose, 0.1% bromophenol blue, 0.1% xylene cyanol loading dye was added immediately prior to native gel electrophoresis. Electrophoresis was carried out at 4°C using 8%, 39:1 (mono:bis) polyacrylamide 1X TH gels (33 mM Tris, 66 mM HEPES, pH ~8) with 5 mM MgCl$_2$ or 2 mM EDTA added as indicated.

## In trans protection experiments

In the experiments presented here, 300 pmol of (+31) or (+0) leadered xrRNAs were reacted with 150 pmol of Xrn1 in 1X EC3 for 30 min at the temperature indicated. At 30 min, 450 pmol of a synthetic, 5′-monophosphorylated 24-mer was added and the reaction continued for 30 min. At 1 hr these reactions were quenched by the addition of an 8 M urea, 10 mM EDTA loading buffer and subjected to 10% dPAGE. Gels were visualized by staining with methylene blue.

## Viral mutagenesis

Mutant Kunjin viruses predicted to have defective sfRNA formation during infection were generated by overlap extension PCR. The FLSDX(pro)HDVr Kunjin virus infectious clone (provided by the Khromykh laboratory; described in *Liu et al., 2003*) was used as a PCR template for the construction of each mutant virus. The high fidelity Phusion Hot Start II polymerase (Thermo Scientific, Pittsburg, PA) was used for PCR using the mutagenic primers listed below and assembled PCR products were inserted into the Age1 and Xho1 restriction sites. Plasmids were screened for the proper mutations by sequencing PCR amplicons using the following primers: FLSDX Fw: 5′- actttgttaattgtaaataaatattgttat; FLSDX Rv: 5′-gcgtgggacgttgattcgcctttgt. Plasmids were linearized with Xho1 and in vitro transcriptions performed using the MEGAscript Sp6 transcription kit (Life Technologies, Foster City, CA) followed by

Turbo DNase treatment to remove template. RNAs were phenol-chloroform-isoamyl alcohol (25:24:1) extracted and ethanol precipitation with ammonium acetate.

## Viral infections

The baby hamster kidney cell line BHK-21 (ATCC CCL-10; *Mesocricetus auratus*) was used to generate Kunjin virus stocks and cells were maintained in MEM plus 10% fetal bovine serum (Atlas Biologicals, Fort Collins, CA) and 1% streptomycin and penicillin (Fisher Scientific-Hyclone, Logan, UT) at 37°C in the presence of 5% $CO_2$. BHK-21 cells were electroporated and virus was amplified by passaging once on BHK-21 cells to generate working stocks. Viral titers were assessed by plaque titrations, infections were performed in human 293T cells (MOI of 10), cells were washed twice after a 2 hr adsorption period, and total cellular RNA was collected at 48 h.p.i using TRIzol (Invitrogen, Carlsbad, CA). Total RNA was treated with DNase I (Fermentas, Vilnius, Lithuania) to remove residual genomic DNA. Northern blotting was performed as described previously (*Moon et al., 2012*) using two micrograms of total RNA from each sample using a probe to the entire 3′ UTR.

## Northern blots

For Kunjin virus sfRNA detection, 2 µg of total RNA from infected cells at 48 h.p.i. (or from mock infected cells) was separated on a 5% denaturing polyacrylamide gel. RNA was then transferred onto a nylon membrane (Hybond-XL; GE Healthcare) for blotting and UV cross-linked before blocking for 30 min at 60°C in hybridization solution (50% formamide, 1 mg/ml bovine serum albumin, 750 mM sodium chloride, 75 mM sodium citrate, 0.1 mg/ml salmon sperm DNA, 1% sodium dodecyl sulfate, 1 mg/ml polyvinylpyrrolidone, 1 mg/ml ficoll). In vitro transcribed, internally radiolabeled RNA probes to the entire 3′ untranslated region of Kunjin virus (GenBank: AY274504.1 nts 10,396-11,022) were generated using a template containing the Kunjin 3′ UTR inserted into the pGEM-4 vector (X65303.1; GenBank) in the EcoRI and HindIII restriction sites using these primers: 5′-GAATT-CTAAATACTTTGTTAATTGTAAAT; 5′-AAGCTTAGATCCTGTGTTCTCGCACCACCA. Following an overnight incubation at 60°C to hybridize the probe to the membrane, blots were washed twice with wash solution (300 mM sodium chloride, 0.1% sodium dodecyl sulfate, 30 mM sodium citrate) and twice with stringent wash solution (30 mM sodium chloride, 0.1% sodium dodecyl sulfate, 3 mM sodium citrate) for 30 min each at 60°C. Hybridized RNAs were visualized by exposing the blot on storage phosphor screens and imaging on the Typhoon Trio Imager (GE Healthcare).

## Mutagenic primers to interrogate the three-way junction in WNV$_{KUN}$

| Mutation/primer set | Forward | Reverse |
|---|---|---|
| Flanking primers | CATACCGGTCGGAAAAGTGATCGACCTTGG | CATCTCGAGCAATTGTTGTTGTTAACTTG |
| G→C10519 | TTCCCGGCACCGGAAGTTGAG | CTCAACTTCCGGTGCCGGGAA |
| G→C10680 | TGGCGTGGCACTCTGCGGAG | CTCCGCAGAGTGCCACGCCA |

## Acknowledgements

The authors thank current and former Kieft Lab members for thoughtful discussions and technical assistance. We thank members of the Das and Yoon Labs at Stanford and Seoul University for their help in implementing HiTRACE software. The expression vectors for BdRppH and KlXrn1 were gifts of Dr Joel Belasco at the Skirball Institute, New York University School of Medicine and Dr Liang Tong at Columbia University, respectively. JSK is an Early Career Scientist of the Howard Hughes Medical Institute.

## Additional information

### Funding

| Funder | Grant reference number | Author |
|---|---|---|
| Howard Hughes Medical Institute | | Jeffrey S Kieft |
| National Institutes of Health | GM081346, GM097333 | Jeffrey S Kieft |

| Funder | Grant reference number | Author |
|---|---|---|
| National Institutes of Health | U54 AI-065357 | Jeffrey Wilusz |

The funders had no role in study design, data collection and interpretation, or the decision to submit the work for publication.

## Author contributions

EGC, Conception and design, Acquisition of data, Analysis and interpretation of data, Drafting or revising the article; JSK, Conception and design, Analysis and interpretation of data, Drafting or revising the article; SLM, Acquisition of data, Analysis and interpretation of data, Drafting or revising the article; JW, Analysis and interpretation of data, Drafting or revising the article

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
