## [Decision Letter]

Thank you for sending your work entitled “RNA structures that resist riboexonuclease Xrn1 produce a pathogenic Dengue virus RNA” for consideration at *eLife*. Your article has been favorably evaluated by a Senior editor and 3 reviewers, one of whom, Timothy Nilsen, is a member of our Board of Reviewing Editors.

The Reviewing editor and the other reviewers discussed their comments before we reached this decision, and the Reviewing editor has assembled the following comments to help you prepare a revised submission.

All agreed that the work was well done and interesting, but we were concerned that the manuscript came up a bit short at present for the expectations of *eLife*. It was agreed that the manuscript would be significantly elevated if it were shown that the three-helix junction structure was important for viral replication of sfRNA levels in a cell-based assay. Alternatively, if a second Xrn1 resistant structure from a different virus was analyzed perhaps in a bit less detail, it would establish the generality or not of the observations you have made. A revised paper must address either of these points.

---

## [Author Response]

We chose to address both of these points. First, we interrogated the Xrn1-resistant properties of two putative Xrn1-resistant RNA structures (xrRNAs) present in the 3’UTR of a flavivirus related to Dengue: the Kunjin strain of West Nile Virus (WNV_KUN_). We found that these RNA structures also resist Xrn1 in vitro and their function demonstrates similar sensitivity to mutations made in conserved sequence elements. These data are contained in Figure 10. Second, because WNV_KUN_ is a convenient strain for cell-based assays, we used it to correlate the observations we made regarding Xrn1 resistance in vitro with the formation of sfRNAs during infection. Specifically, we introduced mutations in the virus that match those we studied in vitro (in the three-way junction), and observed that these decrease sfRNA formation in infected cells. These data are contained in Figure 11. As a result of including these experiments, we added two authors, who have read and fully agree with the manuscript. We believe these experiments establish the generality of our observations and the direct link between the presence of Xrn1-resistant RNA structures and the production of sfRNAs in human cell lines and thus fulfill the requirements for resubmission of our manuscript.